# *VNN1* Gene Expression and Polymorphisms Associated with Chicken Carcass Traits

**DOI:** 10.3390/ani14131888

**Published:** 2024-06-26

**Authors:** Siyu Zhang, Xiang Fang, Ruiquan Wu, Qinghua Nie, Zhenhui Li

**Affiliations:** 1State Key Laboratory of Swine and Poultry Breeding Industry, Guangdong Provincial Key Lab of Agro-Animal Genomics and Molecular Breeding, College of Animal Science, South China Agricultural University, Guangzhou 510642, China; agctu.zhang@stu.scau.edu.cn (S.Z.); nqinghua@scau.edu.cn (Q.N.); 2Key Laboratory of Chicken Genetics, Breeding and Reproduction, Ministry of Agriculture and Rural Affair, South China Agricultural University, Guangzhou 510642, China

**Keywords:** *VNN1*, Vanin-1, polymorphism, carcass trait, association analysis, chicken

## Abstract

**Simple Summary:**

In chicken breeding, direct breeding selection on phenotype is challenging and expensive. Molecular-marker-assisted selection breeding remains an efficient and effective breeding method. Through association analysis, we identified *VNN1* as a significant marker for chicken carcass traits. Single nucleotide polymorphisms in *VNN1* can serve as molecular markers to assist in selection breeding. We propose that these markers can enhance the production efficiency of broilers.

**Abstract:**

This study aimed to investigate the association between hepatic *VNN1* expression and carcass traits in Mahuang chickens as well as to identify polymorphisms in the upstream and downstream regions of *VNN1* that could potentially be associated with these carcass traits. The study revealed that *VNN1* expression levels in liver correlated with various carcass traits such as dressed weight, eviscerated weight, and abdominal fat weight. A total of 39 polymorphic sites were identified, among which 23 were found to be associated with 15 different carcass traits. These polymorphic sites were organized into three distinct haplotype blocks, with BLOCK2 and BLOCK3 being associated with various eviscerated weight percentages, thigh weight, breast muscle weight, wing weight, and other traits. The study underscores the significant role of *VNN1* in influencing the carcass traits of Mahuang chickens and sheds light on the genetic foundations of these traits. The findings provide valuable insights that could inform breeding strategies aimed at optimizing traits relevant to market demands and slaughtering efficiency.

## 1. Introduction

Chicken meat is an important source of protein in the human diet. In China, yellow-feather broilers accounted for 25.21% of the total chicken meat yield in 2023 [1]. Breeding efforts for yellow-feather broilers have traditionally focused on body weight and feather color, leading to suboptimal carcass traits. To meet the demands of segmented sales, improving carcass traits has become an important goal in the breeding of yellow-feather broilers.

Carcass traits in chickens, which encompass characteristics such as live weight (LW), dressed weight (DW), eviscerated weight (EW), eviscerated weight with giblet (EWG), breast muscle weight (BMW), thigh weight (TW), wing weight (WW), and abdominal fat weight (AFW), are of critical importance for production and are closely linked to the economic performance of the broilers. In recent years, genome-wide association studies (GWASs) have been utilized to identify the quantitative trait loci (QTLs) and genes associated with carcass traits [2]. However, QTLs and related genes associated with carcass traits often do not overlap across different breed populations [3,4,5,6], making these markers non-generalizable. Additionally, genomic selection breeding is currently expensive, whereas marker-assisted selection (MAS) remains a cost-effective and rapid breeding method [7].

Mahuang chickens are a renowned yellow-feather broiler breed for their growth performance and meat quality. There is a lack of markers for carcass traits in Mahuang chicken breeding. The liver serves as a metabolic central organ that interconnects to various tissues, including skeletal muscle and adipose tissue, thereby controlling the energy metabolism of the whole body [8]. A key gene involved in liver metabolism is *VNN1*, which encodes for pantetheine hydrolase (Vanin-1) and is predominantly expressed in the liver [9]. Previous research has demonstrated that *VNN1* is regulated by PPARα and influences lipid and glucose metabolism in the liver [10]. Moreover, gain- and loss-of-function studies have revealed that *VNN1* is activated by the synergistic interaction of peroxisome proliferator-activated receptor γ coactivator 1α (PGC-1α) and hepatocyte nuclear factor-4α (HNF4α), which subsequently induce the expression of phosphoenolpyruvate carboxykinases (PEPCK) and glucose 6-phosphatase (G6Pase), activating hepatic gluconeogenesis [11]. In chickens, *VNN1* is regulated by PPARα and is speculated to participate in liver lipid metabolism [12,13,14]. However, further studies to confirm whether *VNN1* is related to carcass traits are still lacking.

In this study, the carcass traits included were statistically determined in 432 Mahuang chickens, the liver *VNN1* expression levels were detected, and SNPs in the regions flanking the *VNN1* gene were screened. The aim was to analyze the relationship between carcass traits and hepatic *VNN1* expression, and to find SNP markers within *VNN1* associated with these carcass traits.

## 2. Materials and Methods

### 2.1. Carcass Traits Data

The South China Agricultural University Institutional Animal Care and Use Committee approved all sampling and laboratory procedures adopted in this study (approval ID: SCAU#2021F074).

A total of 432 Mahuang chickens of same breeding line were the animals used in this study. After hatching, meconium avian leukosis was checked by using a double antibody sandwich enzyme-linked immunosorbent assay kit (P27 antigen), and positive individuals were eliminated. Individuals with an initial body weight of less than 42 g were eliminated. Qualified chickens were raised according to the standard feeding and management procedures by KwangFeng Industrial Co., Ltd. (Guangzhou, China). All the chickens were kept in a brooder house until 6 weeks of age and were then moved to the grower house with single-cage rearing. All of the chickens were reared in stepped cages [15] in the same pen under intensive management providing the same management regimen with ad libitum feed (Table 1) and watering until 90 days of age.

All 90-day-old chickens were humanely slaughtered by cervical dislocation, blood drained from carotid artery, and tissues dissected. The carcass traits including LW, DW, EW, EWG, BMW, TW, WW, AFW, and their percentages (DWP, EWP, EWGP, BMWP, TWP, WWP, and AFWP) were measured and calculated as described by Yang et al. [16].

### 2.2. Primer Design

Primers were designed with Primer-Blast (https://www.ncbi.nlm.nih.gov/tools/primer-blast/index.cgi?LINK_LOC=BlastHome, accessed on 10 March 2022) according to the *VNN1* gene sequence (ENSGALG00000013993), as shown in Table 2.

### 2.3. DNA Extraction, Polymerase Chain Reaction, and DNA Sequencing

All of the blood of Mahuang chickens was collected during slaughtering. DNA was extracted from the blood using E.Z.N.A.^®^ NRBC Blood DNA Kit (OMEGA, Norcross, GA, USA) according to the protocol manual. DNA quality was checked using NANODROP ONE (Thermo Scientific, Waltham, MA, USA), with the criterion of 1.8 < A260/A280 < 2.0, A260/A230 > 2.0, and concentration > 50 ng/μL. A total of 432 DNA samples that met these criteria were used to amplify the dsDNA of *VNN1* fragments with 2× Taq MasterMix (CWBIO, Nanjing, China). PCR conditions were 94 °C for 2 min, followed by cycles of 94 °C for 30 s, 60 °C for 30 s, 72 °C for 30 s, and a final extension of 72 °C for 5 min. Finally, Sanger sequencing of the PCR products was carried out by Guangzhou Tianyi Huiyuan Gene Technology Co., Ltd. (Guangzhou, China).

### 2.4. RNA Extraction, Complementary DNA (cDNA) Synthesis, and Quantitative Real-Time PCR (qRT-PCR)

For RNA extraction, liver tissues were collected from 47 female Mahuang chickens randomly. The tissues were immediately placed into cryovials and frozen inside liquid nitrogen during slaughtering. Afterward, they were stored at −80 °C.

Total RNA was isolated from livers using RNA isolator Total RNA Extraction Reagent (Vazyme, Nanjing, China) following the manufacturer’s instructions. A total of 200 μL cDNA of approximately 800 ng RNA per sample was synthesized with MonScript™ RTIII All-in-One Mix with dsDNase (Monad, Wuhan, China). The qRT-PCR program (95 °C for 30 s, followed by cycles of 95 °C for 10 s, 60 °C for 30 s) was carried out in a Bio-Rad CFX96 system (Bio-Rad, Hercules, CA, USA) with ChamQ SYBR qPCR Master Mix (Vazyme, Nanjing, China) with 5 μL ChamQ SYBR qPCR Master Mix, 4.2 μL cDNA, and 8 μL primer (10 μM) used for each reaction. Chicken GAPDH was used as the internal control. The relative expression of genes was analyzed with the comparative 2^−∆∆Ct^ method [17].

### 2.5. Sequencing and Genotyping

For quality control, sequence reads shorter than 700 bp were filtered out. Qualified reads were then examined using the SnapGene software suite (version 4.3.6) to assess the chromatograms. The criteria for qualified reads included the following:

Peak Heights: reads with peak heights of 500 or more for most bases.

Peak Masses: reads with peak masses of 50 or more for most bases.

Finally, the first 30 bp and the last 30 bp of each read were trimmed to remove low-quality bases commonly found at the ends of sequencing reads.

The sequence reads were aligned with the *VNN1* sequence (ENSGALG00000013993) by using SnapGene software suite (version 4.3.6) for genotyping. We genotyped SNPs according to Yang et al. [16].

### 2.6. Statistics and Analysis

The phenotype data was estimated with the program R (version 4.1.2). The allele frequency, genotype frequency, and Hardy–Weinberg disequilibrium (HWE) were calculated with PLINK 1.9 [18,19]. The polymorphism information content (PIC) [20] was calculated with Gene-Calc (https://gene-calc.pl/, accessed on 4 November 2022). The linkage disequilibrium and haplotypes were determined and visualized with the gaston package in R (version 1.5.7) [21].

Correlation analysis between *VNN1* mRNA expression and carcass traits fitted a simple linear regression model created using Graphpad prism (version 9.5). Association analysis between SNPs (or haplotypes) and carcass traits was carried out with a general linear model (GLM) combined variable using the linear command in PLINK 1.9. The GLM model used was as follows:Y = μ + G + S + e,
where Y is the phenotype value of carcass traits, μ is the population means, G is the fixed effect of the genotype (or haplotype), S is the fixed effect of sex, and e represents the random error.

Analysis of Variance (ANOVA) was performed to estimate the phenotypic differences between genotypes (or haplotypes) in R software (version 4.2.1).

## 3. Results

### 3.1. Information on Carcass Traits

To comprehensively understand the phenotypic information of the 432 (342 female and 89 male) investigated Mahuang chickens, we counted the mean, maximum, minimum, and standard deviation of the traits LW, DW, EW, EWG, BMW, TW, WW, AFW, DWP, EWP, EWGP, BMWP, TWP, WWP, and AFWP by sex. We found significant differences in these traits in both the females and males (Table 3). We then performed Pearson correlation analyses for carcass traits in the male and female chickens. The results show that LW was significantly positive correlated to DW, EW, EWG, BMW, TW, WW, and AFW, and in addition, LW was significantly negatively correlated with WWP (Figure 1). Interestingly, we found that AFWP was negatively correlated with EWGP, BMW (BMWP), TW (TWP), and WW (WWP), and was particularly significant in the females (Figure 1). These findings provide important insights into the relationships between different carcass traits in Mahuang chickens and can inform future breeding and selection strategies for improved meat quality.

### 3.2. Association between VNN1 Expression and Carcass Traits

Given the pivotal role of *VNN1* in hepatic lipid metabolism, we investigated the association between hepatic *VNN1* expression and carcass traits in chickens. We quantified the relative expression level of *VNN1* using qPCR and performed a simple linear regression with the carcass traits. Our results demonstrated a significant association between hepatic *VNN1* expression level and several carcass traits, including DW, DWP, EW, EWP, EWG, AFW, and AFWP (*p* < 0.05, Figure 2A–G), and a non-significant association with LW, EWGP, TW, TWP, WW, and WWP (*p* > 0.05, Appendix A–H). The correlation coefficients (r2) of AFW and AFWP phenotype with *VNN1* expression levels were 0.2337 and 0.1961, respectively (Figure 2F,G). These findings suggest a strong association between hepatic *VNN1* expression levels and abdominal fat deposition.

### 3.3. Information on SNPs

To explore genetic variation in *VNN1*, we identified 39 polymorphic sites, including 19 upstream and 20 downstream SNPs, in the Mahuang chicken population (Figure 3). Among these SNPs, eight were newly identified (C56889810T, C56889913G, T56890010C, G56890028A, C56890154T, A56906727T, A56907254G, T56907387C), while the others can already be found in the Genetic Variation of Ensembl. We found that one SNP was a deletion mutation, and two SNPs were insertion mutations, while the others were base conversion mutations (Table 4). Table 5 shows the genotype and allelic frequencies, Hardy–Weinberg disequilibrium (HWE), and polymorphism information content (PIC) for each SNP. Our results suggest that the SNPs in the downstream region deviated from the HWE (*p* < 0.05), possibly due to selection during breeding. The PIC ranged from 0.054 to 0.519, indicating low or moderate levels of polymorphism, with the exception of rs734452255, which exhibited high polymorphism (PIC > 0.5).

### 3.4. Association of SNPs with Carcass Traits and Haplotype Reconstruction

We analyzed the association between carcass traits and 39 SNPs using GLM, with a significance threshold of *p* < 0.05. Of these SNPs, 23 were significantly associated with carcass traits (Figure 4A, Appendix A). ANOVA was performed to estimate differences in AFW between genotypes at rs736763713. Chickens with the AA genotype had significantly higher AFW compared to those with GA or GG genotypes (*p* < 0.05, Appendix A).

To investigate the multi-loci association between the haplotype structures of the SNPs and carcass traits, we constructed linkage disequilibrium (LD) blocks. We defined three blocks from SNPs associated with carcass traits, with r2 measures of LD ranging from 0 to 1 (Figure 4B–E and Appendix A). Haplotypes with frequencies > 0.01 and diplotypes with frequencies > 0.005 were reserved and reconstructed based on population genotype data, which are shown in Table 6.

### 3.5. Association of Haplotypes with Carcass Traits

To comprehensively elucidate the relationship between SNPs and carcass traits, the LD blocks obtained from linkage disequilibrium were analyzed using the GLM with 15 carcass traits. As indicated in Appendix A, BLOCK1 was not associated with any traits, while BLOCK2 was related to EW, EWP, EWGP, TW, and TWP, and BLOCK3 was associated with EW, EWP, EWG, EWGP, BMW, BMWP, TW, TWP, WW, and WWP. Multiple comparisons of phenotypic differences between the diplotypes were performed using ANOVA. No significant phenotypic differences were observed between the diplotypes of BLOCK2 and BLOCK3 (Table 7, Table 8, and Appendix A). However, the H1H2 diplotype of BLOCK2 was advantageous for EW, EWP, EWGP, and TW, as shown in Table 7. In Table 8, the H2H2 diplotype demonstrated higher values of EW, EWP, EWG, BMW, TW, and WW than other diplotypes.

## 4. Discussion

Here, we present an association analysis of carcass traits and *VNN1* gene in Mahuang chickens. Our analysis reveals a relationship between hepatic *VNN1* expression and several carcass traits, including DW, DWP, EW, EWP, EWG, AFW, and AFWG. Additionally, we found associations between SNPs located upstream and downstream of the *VNN1* gene and these carcass traits. The upstream SNPs, including rs738527022, rs315414930, rs732101949, rs735639580, rs317230140, rs317648691, and rs315560057, form a haplotype (BLOCK2) that is associated with EW, EWP, EWGP, TW, and TWP. The downstream SNPs, including rs730937100, rs314708583, rs735640639, rs739066967, rs731721017, and rs732782882, form another haplotype (BLOCK3) that is associated with EW, EWP, EWG, EWGP, BMW, BMWP, TW, TWP, WW, and WWP.

Our findings demonstrate that hepatic *VNN1* expression levels are associated with various carcass traits such as DW, DWP, EW, EWP, EWG, AFW, and AFWP, and especially AFW and AFWP, with the latter showing the strongest correlation. There is a high correlation between DW and carcass performance (EW and EWG). These results suggest that *VNN1* may affects carcass traits. Carcass traits have been studied by other authors and associated with gene expression and gene polymorphisms [22,23,24,25,26,27]. Identifying SNPs of VNN1 could be a potential way to find molecular markers associated with carcass traits, thereby facilitating faster genetic gain in breeding programs.

The upstream and downstream regions of genes are often rich in elements that influence transcription and post-transcription regulation, such as transcription factor binding sites, miRNA binding sites, and upstream open reading frames (uORFs) [28,29,30]. Prior research has linked polymorphisms in these regions to variations in gene expression and growth performance [31,32,33,34,35]. In our study, we screened the *VNN1* flanking regions and identified 39 SNPs, with 23 linked to carcass traits and 8 reported for the first time. Notably, we constructed three haplotype blocks, with BLOCK2 and BLOCK3 showing strong associations with carcass traits. These blocks offer potent markers for breeding strategies.

Genetic variation occurs continuously throughout the genome. In chickens, different breeds are subjected to varying selection pressures, resulting in differences in genetic structure and SNP distribution. The SNPs and haplotypes associated with the carcass traits in Mahuang populations identified in this study may not be directly applicable to other breeds. However, among the 39 SNPs examined, only 8 were novel to Mahuang chickens, and these novel SNPs were not significantly associated with carcass traits. Interestingly, the SNPs found to be associated with carcass traits in Mahuang chickens have also been identified in other breeds (named by Ensembl), suggesting the potential for broader application of these SNP markers across different chicken populations.

## 5. Conclusions

Our research identified potential effects of hepatic *VNN1* expression on carcass traits and identified SNPs and haplotype blocks on the *VNN1* gene, providing promising markers for improving carcass traits in Mahuang chicken breeding.

## Figures and Tables

**Figure 1 animals-14-01888-f001:**
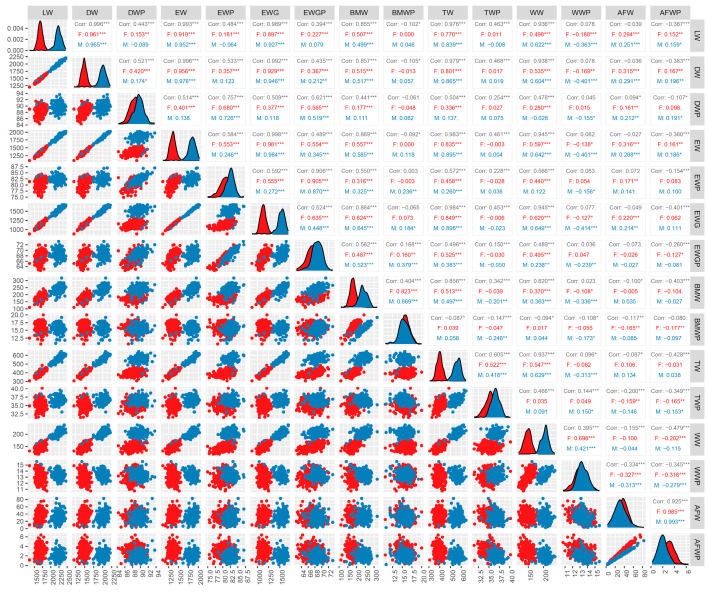
Correlation between carcass traits in Mahuang chickens. Abbreviations: LW = live weight; DW = dressed weight; DWP = dressed weight percentage; EW = eviscerated weight; EWP = eviscerated weight percentage; EWG = eviscerated with giblet; EWGP = eviscerated with giblet percentage; BMW = breast muscle weight; BMWP = breast muscle weight percentage; TW = thigh weight; TWP = thigh weight percentage; WW = wing weight; WWP = wing weight percentage; AFW = abdominal fat weight; AFWP = abdominal fat weight percentage; F (red): female; M (blue): male. “***” if the *p*-value is <0.001; “**” if the *p*-value is <0.01; “*” if the *p*-value is <0.05.

**Figure 2 animals-14-01888-f002:**
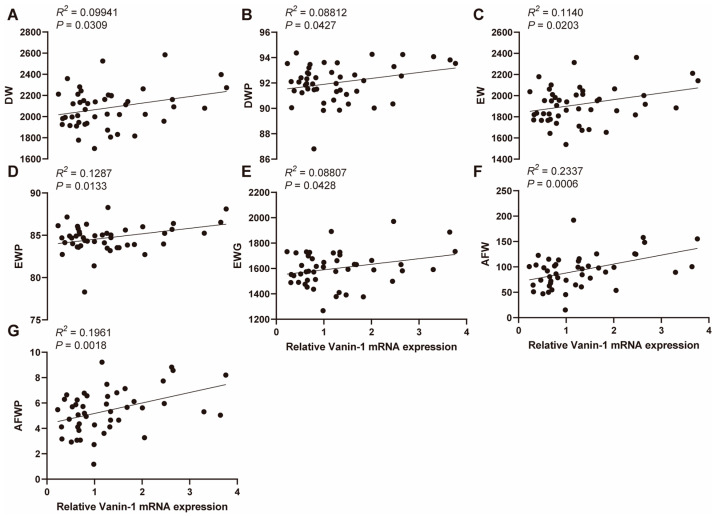
Association between liver *VNN1* mRNA expression and carcass traits (**A**–**G**). Association between liver *VNN1* mRNA expression and DW, DWP, EW, EWP, EWG, AFW, and AFWP in 47 female Mahuang chickens. Abbreviations: DW = dressed weight; DWP = dressed weight percentage; EW = eviscerated weight; EW = eviscerated weight percentage; EWG = eviscerated with giblet; AFW = abdominal fat weight; AFWP = abdominal fat weight percentage.

**Figure 3 animals-14-01888-f003:**
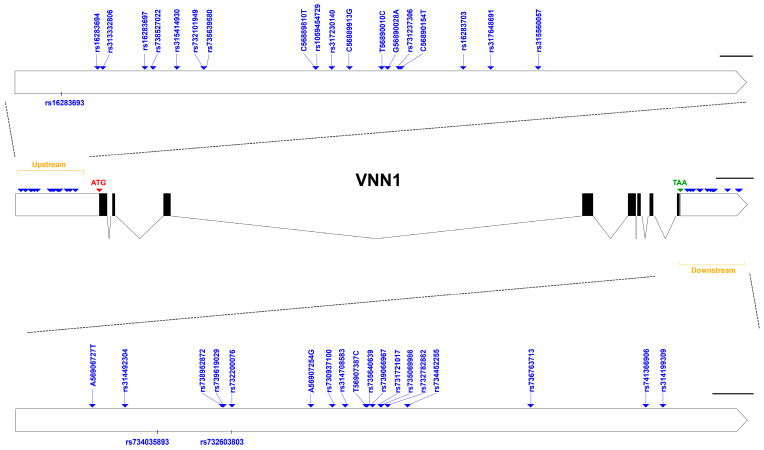
Location distribution of SNPs on *VNN1* gene.

**Figure 4 animals-14-01888-f004:**
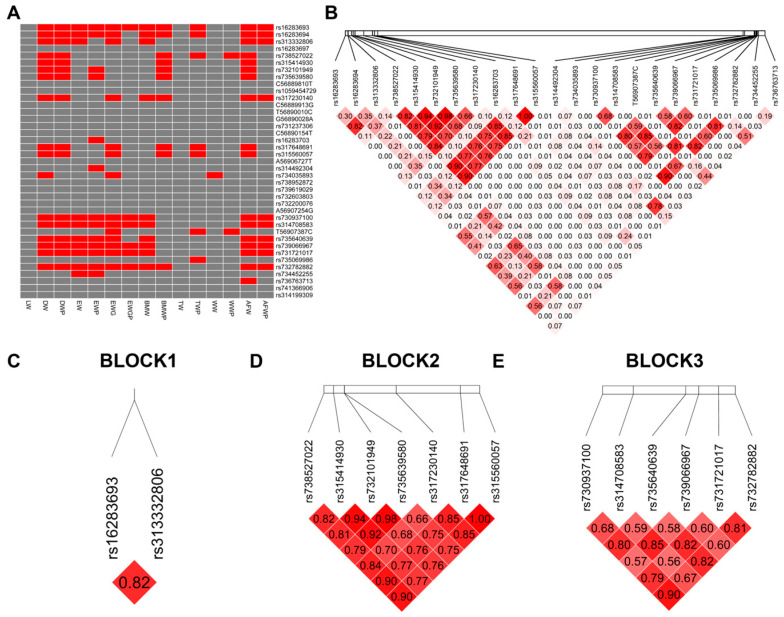
LD plot shows the linkage status of 23 SNPs associated with carcass traits in the *VNN1* gene. (**A**) Association analysis of the SNPs with carcass traits in chickens. “Red” indicates a *p* value less than 0.05, while “grey” indicates a *p* value more than 0.05. (**B**) The linkages between 23 SNPs. (**C**–**E**) Three blocks constructed from 23 SNPs. The color of the block indicates the LD status of SNPs; deep red means high linkages between SNPs. The number of the block shows the linkages between SNPs in r2. Abbreviations: LW = live weight; DW = dressed weight; DWP = dressed weight percentage; EW = eviscerated weight; EWP = eviscerated weight percentage; EWG = eviscerated with giblet; EWGP = eviscerated with giblet percentage; BMW = breast muscle weight; BMWP = breast muscle weight percentage; TW = thigh weight; TWP = thigh weight percentage; WW = wing weight; WWP = wing weight percentage; AFW = abdominal fat weight; AFWP = abdominal fat weight percentage.

**Table 1 animals-14-01888-t001:** Nutrition during feeding.

Nutrient	1–6 Weeks	7–13 Weeks
Energy Metabolism (kcal/kg)	2870	2760
Crude protein (%)	17.4	15.6
Calcium (%)	1.02	1.10
Phosphorus (%)	0.55	0.60
Lysine (%)	1.0	0.8
Methionine (%)	0.46	0.38
Sodium chloride (%)	0.43	0.44

**Table 2 animals-14-01888-t002:** Details of primers used in this study.

Primer Name	Sequence (5′-3′)	Purpose	Length (bp)
VNN1-SNP5	F: GGAGGCTACTTGATCTTCCTGG	Amplification	1746
	R: CTCCCCAGTTTACCCTCCCTA		
VNN1-SNP5-S	F: GTAAGTAATCATGTAAAT	Sequencing	-
VNN1-SNP3	F: CCCTGCGAAGCTTCCTGTAA	Amplification	1712
	R: GCTGCTCTCAGCTGCTCTTA		
VNN1-SNP3-S	F: TTGGGACTCCTAGTGAAG	Sequencing	-
VNN1	F: CTGCACCGATCCCACAAGAT	qPCR	146
	R: GACCATCACTGGGGCACTTG		
GAPDH	F: AGAACATCATCCCAGCGT	qPCR	182
	R: AGCCTTCACTACCCTCTTG		

Primers: VNN1-SNP5 and VNN1-SNP3 were used for amplification and detection of SNPs. VNN1-SNP5-S and VNN1-SNP3-S were used for genotyping. VNN1 and GAPDH were used for qRT-PCR.

**Table 3 animals-14-01888-t003:** Descriptive statistics of carcass traits.

Trait	Mean ± SD		Maximum		Minimum	
	Female	Male	Female	Male	Female	Male
LW	1679.97 ± 75.16 ^a^	2217.08 ± 120.63 ^b^	1890.00	2430.00	1350.00	1570.00
DW	1484.26 ± 72.47 ^a^	1989.84 ± 108.28 ^b^	1699.00	2196.00	1202.00	1421.00
DWP	88.34 ± 1.22 ^a^	89.76 ± 1.36 ^b^	91.74	93.66	84.34	86.13
EW	1355.71 ± 71.89 ^a^	1828.76 ± 102.16 ^b^	1562.00	1995.00	1099.00	1309.00
EWP	80.68 ± 1.72 ^a^	82.49 ± 1.38 ^b^	84.48	87.00	74.91	78.41
EWG	1131.84 ± 64.02 ^a^	1520.7 ± 93.35 ^b^	1320.00	1681.00	880.00	1040.00
EWGP	67.36 ± 1.74 ^a^	72.23 ± 4.28 ^b^	71.99	79.63	62.55	64.90
BMW	178.74 ± 17.75 ^a^	233.553 ± 26.21 ^b^	242.49	318.00	110.00	141.26
BMWP	15.79 ± 1.23 ^a^	15.34 ± 1.24 ^b^	19.99	19.18	10.62	11.84
TW	398.72 ± 26.28 ^a^	556.99 ± 37.33 ^b^	485.34	625.65	307.00	390.8
TWP	35.23 ± 1.23 ^a^	36.63 ± 1.01 ^b^	40.18	38.69	31.45	34.46
WW	148.21 ± 11.6 ^a^	203.51 ± 12.18 ^b^	178.41	228.23	118.00	162.01
WWP	13.1 ± 0.82 ^a^	13.4 ± 0.71 ^b^	15.12	15.58	10.73	11.86
AFW	35.38 ± 11.79 ^a^	30.84 ± 13.36 ^b^	76.16	83.00	4.00	5.29
AFWP	3.02 ± 0.96 ^a^	1.98 ± 0.83 ^b^	6.39	5.20	0.36	0.33

T-test was performed to estimate the phenotypic differences between female and male. Values differ significantly at *p* < 0.05 with different superscripts. Units: g. Abbreviations: LW = live weight; LWP = live weight percentage; DW = dressed weight; DWP = dressed weight percentage; EW = eviscerated weight; EWP = eviscerated weight percentage; EWG = eviscerated with giblet; EWGP = eviscerated with giblet percentage; BMW = breast muscle weight; BMWP = breast muscle weight percentage; TW = thigh weight; TWP = thigh weight percentage; WW = wing weight; WW = wing weight percentage; AFW = abdominal fat weight; AFWP = abdominal fat weight percentage.

**Table 4 animals-14-01888-t004:** SNPs identified in the *VNN1* gene.

SNP Name	Chr: bp ^1^	Alleles ^2^	Class ^3^	Conseq.Type ^4^
rs16283693	3:56889037	A/-	deletion	upstream gene variant
rs16283694	3:56889150	A/G	SNP	upstream gene variant
rs313332806	3:56889166	A/C	SNP	upstream gene variant
rs16283697	3:56889293	A/G	SNP	upstream gene variant
rs738527022	3:56889318	A/G	SNP	upstream gene variant
rs315414930	3:56889390	C/T	SNP	upstream gene variant
rs732101949	3:56889471	A/C	SNP	upstream gene variant
rs735639580	3:56889472	A/G	SNP	upstream gene variant
C56889810T	3:56889810	C/T	SNP	upstream gene variant
rs1059454729	3:56889811	G/A	SNP	upstream gene variant
rs317230140	3:56889859	G/A	SNP	upstream gene variant
C56889913G	3:56889913	C/G	SNP	upstream gene variant
T56890010C	3:56890010	T/C	SNP	upstream gene variant
G56890028A	3:56890028	G/A	SNP	upstream gene variant
rs731237306	3:56890062	A/G	SNP	upstream gene variant
C56890154T	3:56890154	C/T	SNP	upstream gene variant
rs16283703	3:56890257	T/G	SNP	upstream gene variant
rs317648691	3:56890340	C/A	SNP	upstream gene variant
rs315560057	3:56890484	A/G	SNP	upstream gene variant
A56906727T	3:56906727	A/T	SNP	3 prime UTR variant
rs314492304	3:56906806	T/A	SNP	3 prime UTR variant
rs734035893	3:56906872-56906873	-/C	insertion	3 prime UTR variant
rs738952872	3:56907040	G/A	SNP	3 prime UTR variant
rs739619029	3:56907043	T/A	SNP	3 prime UTR variant
rs732603803	3:56907060-56907061	-/C	insertion	3 prime UTR variant
rs732200076	3:56907063	G/A	SNP	3 prime UTR variant
A56907254G	3:56907254	A/G	SNP	3 prime UTR variant
rs730937100	3:56907306	C/A	SNP	3 prime UTR variant
rs314708583	3:56907337	G/A	SNP	3 prime UTR variant
T56907387C	3:56907387	T/C	SNP	3 prime UTR variant
rs735640639	3:56907390	G/C	SNP	3 prime UTR variant
rs739066967	3:56907403	A/T	SNP	3 prime UTR variant
rs731721017	3:56907423	C/T	SNP	3 prime UTR variant
rs735069986	3:56907424	G/A	SNP	3 prime UTR variant
rs732782882	3:56907440	T/C	SNP	3 prime UTR variant
rs734452255	3:56907488	T/C	SNP	3 prime UTR variant
rs736763713	3:56907786	G/A	SNP	3 prime UTR variant
rs741366906	3:56908065	A/C	SNP	3 prime UTR variant
rs314199309	3:56908106	G/A	SNP	3 prime UTR variant

^1^ Chr:bp indicates which chromosome and which position the SNP is located in. ^2^ Alleles show alternative nucleotides. ^3^ Class means mutation type of SNP. ^4^ Conseq.Type means what consequence type will cause SNP.

**Table 5 animals-14-01888-t005:** Genotype frequencies, allelic frequencies, and HWE of SNPs.

SNP Name	N	Genotype Frequency			Allelic Frequency	Ho	He	*p* Value	PIC
rs16283693	424	AA(0.002)	A-(0.087)	--(0.910)	A(0.046)	0.087	0.088	0.600	0.151
rs16283694	424	AA(0.007)	AG(0.215)	GG(0.778)	A(0.114)	0.215	0.203	0.335	0.292
rs313332806	424	AA(0.002)	AC(0.085)	CC(0.913)	A(0.045)	0.085	0.086	0.580	0.148
rs16283697	424	AA(0.033)	AG(0.292)	GG(0.675)	G(0.179)	0.293	0.294	0.869	0.379
rs738527022	420	AA(0.052)	AG(0.364)	GG(0.583)	A(0.235)	0.364	0.359	0.892	0.431
rs315414930	389	CC(0.041)	CT(0.316)	TT(0.643)	C(0.199)	0.316	0.319	0.874	0.401
rs732101949	389	AA(0.039)	AC(0.324)	CC(0.638)	A(0.201)	0.324	0.321	1.000	0.400
rs735639580	389	AA(0.039)	AG(0.321)	GG(0.64)	A(0.199)	0.321	0.319	1.000	0.399
C56889810T	424	CC(0.005)	CT(0.134)	TT(0.861)	C(0.072)	0.134	0.134	1.000	0.214
rs1059454729	424	GG(0.002)	GA(0.134)	AA(0.863)	A(0.070)	0.134	0.130	0.710	0.210
rs317230140	426	GG(0.049)	GA(0.343)	AA(0.608)	G(0.221)	0.343	0.344	1.000	0.421
C56889913G	425	CC(0.014)	CG(0.202)	GG(0.784)	G(0.115)	0.202	0.204	0.812	0.294
T56890010C	425	TT(0.014)	TC(0.198)	CC(0.788)	C(0.113)	0.198	0.200	0.807	0.291
G56890028A	425	GG(0.005)	GA(0.096)	AA(0.899)	A(0.053)	0.096	0.100	0.328	0.168
rs731237306	425	AA(0.014)	AG(0.191)	GG(0.795)	G(0.109)	0.191	0.195	0.618	0.285
C56890154T	425	CC(0.012)	CT(0.188)	TT(0.800)	T(0.106)	0.188	0.189	0.800	0.279
rs16283703	425	TT(0.068)	TG(0.442)	GG(0.489)	G(0.289)	0.442	0.411	0.156	0.462
rs317648691	425	CC(0.054)	CA(0.365)	AA(0.581)	C(0.237)	0.365	0.361	0.894	0.434
rs315560057	425	AA(0.054)	AG(0.365)	GG(0.581)	A(0.237)	0.365	0.361	0.894	0.434
A56906727T	393	AA(0.018)	AT(0.023)	TT(0.959)	T(0.029)	0.023	0.057	6.01 × 10^−10^	0.077
rs314492304	393	TT(0.214)	TA(0.188)	AA(0.598)	T(0.308)	0.188	0.426	8.19 × 10^−28^	0.500
rs734035893	393	--(0.135)	-C(0.036)	CC(0.83)	C(0.153)	0.036	0.259	2.58 × 10^−50^	0.266
rs738952872	393	GG(0.015)	GA(0.018)	AA(0.967)	A(0.024)	0.018	0.047	2.27 × 10^−09^	0.064
rs739619029	393	TT(0.015)	TA(0.018)	AA(0.967)	A(0.024)	0.018	0.047	2.27 × 10^−09^	0.064
rs732603803	393	--(0.018)	-C(0.01)	CC(0.972)	C(0.023)	0.010	0.045	2.36 × 10^−12^	0.054
rs732200076	393	GG(0.018)	GA(0.01)	AA(0.972)	A(0.023)	0.010	0.045	2.36 × 10^−12^	0.054
A56907254G	393	AA(0.104)	AG(0.015)	GG(0.88)	G(0.112)	0.015	0.199	6.11 × 10^−49^	0.197
rs730937100	393	CC(0.02)	CA(0.056)	AA(0.924)	A(0.048)	0.056	0.092	2.56 × 10^−07^	0.137
rs314708583	393	GG(0.033)	GA(0.056)	AA(0.911)	A(0.061)	0.056	0.115	4.61 × 10^−12^	0.159
T56907387C	393	TT(0.135)	TC(0.003)	CC(0.863)	C(0.136)	0.003	0.235	4.71 × 10^−66^	0.211
rs735640639	393	GG(0.033)	GC(0.048)	CC(0.919)	C(0.057)	0.048	0.108	4.43 × 10^−13^	0.147
rs739066967	393	AA(0.038)	AT(0.053)	TT(0.908)	T(0.065)	0.053	0.121	2.68 × 10^−14^	0.163
rs731721017	393	CC(0.02)	CT(0.053)	TT(0.926)	T(0.047)	0.053	0.090	1.52 × 10^−07^	0.133
rs735069986	392	GG(0.112)	GA(0.013)	AA(0.875)	A(0.119)	0.013	0.209	1.18 × 10^−52^	0.202
rs732782882	393	TT(0.02)	TC(0.053)	CC(0.926)	C(0.047)	0.053	0.090	1.52 × 10^−07^	0.133
rs734452255	385	TT(0.26)	TC(0.179)	CC(0.561)	T(0.349)	0.179	0.455	5.46 × 10^−33^	0.519
rs736763713	385	GG(0.075)	GA(0.179)	AA(0.745)	A(0.165)	0.179	0.276	6.67 × 10^−10^	0.415
rs741366906	385	AA(0.068)	AC(0.049)	CC(0.883)	C(0.092)	0.049	0.167	4.48 × 10^−25^	0.398
rs314199309	385	GG(0.075)	GA(0.036)	AA(0.888)	A(0.094)	0.036	0.170	7.83 × 10^−31^	0.193

Abbreviations: N = number of individuals used in the analysis; Ho = observed frequency of heterozygosity; He: expected frequency of heterozygosity; *p* value: χ^2^ test of Hardy–Weinberg equilibrium; PIC: polymorphism information content, where PIC > 0.5 means high polymorphism, 0.25 < PIC 0.5 means medium polymorphism, and PIC < 0.25 means low polymorphism.

**Table 6 animals-14-01888-t006:** The information of haplotypes.

LD Block ^1^	SNP ^2^	Haplotype ^3^	Diplotype ^4^
BLOCK1	rs16283693 rs313332806	H1: -C(0.952) H2: AA(0.042)	H1H1(0.908) H1H2(0.085)
BLOCK2	rs738527022 rs315414930 rs732101949 rs735639580 rs317230140 rs317648691 rs315560057	H1: GTCGAAG(0.790) H2: ACAAGCA(0.177) H3: ACAAACA(0.016)	H1H1(0.618) H1H2(0.278) H2H2(0.036) H1H3(0.026)
BLOCK3	rs730937100 rs314708583 rs735640639 rs739066967 rs731721017 rs732782882	H1: CGGACT(0.925) H2: AACTTC(0.041) H3: CAGTCT(0.014)	H1H1(0.901) H1H2(0.043) H2H2(0.018) H3H3(0.013)

^1^ LD block shows LD block names that represent the haplotype blocks constructed by SNPs. ^2^ SNP displays SNPs that involved the construction of haplotype blocks. ^3^ Haplotype gives the representative names of the haplotype, the composition of genotypes, and haplotype frequency. ^4^ Diplotype gives the diplotypes of haplotype blocks and their frequencies.

**Table 7 animals-14-01888-t007:** Association analysis of BLOCK2 with carcass traits in F2 generation partridge chickens.

Traits ^1^	BLOCK2 ^2^				*p* Value ^3^
	H1H1	H1H2	H1H3	H2H2	
EW	1455.26 ± 211.68 ^a^	1465.44 ± 212.41 ^a^	1411 ± 176.44 ^a^	1451.23 ± 203.85 ^a^	0.0300
EWP	80.02 ± 1.88 ^a^	81.25 ± 1.84 ^a^	81.06 ± 1.39 ^a^	80.45 ± 1.81 ^a^	0.0086
EWGP	68.28 ± 3.25 ^a^	68.79 ± 3.31 ^a^	68.37 ± 3.20 ^a^	67.35 ± 2.72 ^a^	0.0286
TW	432.43 ± 70.43 ^a^	433.81 ± 74.14 ^a^	420.41 ± 65.31 ^a^	426.15 ± 68.1 ^a^	0.0103
TWP	35.61 ± 1.25 ^a^	35.32 ± 1.42 ^a^	35.63 ± 1.54 ^a^	35.27 ± 1.41 ^a^	2.88 × 10^−4^

^1^ Traits associated with BLOCK2. ^2^ Displaying mean ± SD of traits by diplotype of BLOCK2. ^3^ Showing the *p* value of association between traits and BLOCK2. *p* values less than 0.001 are shown in scientific notation. ^a^ Values within a row with different superscripts differ significantly at *p* < 0.05.

**Table 8 animals-14-01888-t008:** Association analysis of BLOCK3 with carcass traits in F2 generation partridge chickens.

Traits ^1^	BLOCK3 ^2^				*p* Value ^3^
	H1H1	H1H2	H2H2	H3H3	
EW	1421.95 ± 181.59 ^a^	1464.29 ± 201.26 ^a^	1533.5 ± 287.7 ^a^	1331 ± 26.25 ^a^	4.24 × 10^−6^
EWP	80.89 ± 1.80 ^a^	81.47 ± 1.25 ^a^	81.74 ± 1.58 ^a^	80.2 ± 1.15 ^a^	2.14 × 10^−6^
EWG	1186.09 ± 151.01 ^a^	1216.94 ± 169.15 ^a^	1277.67 ± 239.74 ^a^	1113.6 ± 27.08 ^a^	6.02 × 10^−6^
EWGP	67.93 ± 2.74 ^a^	68.8 ± 3.48 ^a^	68.11 ± 1.46 ^a^	67.1 ± 1.47 ^a^	1.62 × 10^−6^
BMW	186.1 ± 26.85 ^a^	191.46 ± 28.28 ^a^	193.2 ± 38.67 ^a^	179.01 ± 10.21 ^a^	0.0007
BMWP	15.7 ± 1.26 ^a^	15.75 ± 1.2 ^a^	15.18 ± 2.00 ^a^	16.07 ± 0.80 ^a^	0.0014
TW	420.89 ± 61.61 ^a^	436.99 ± 70.89 ^a^	457.14 ± 107.21 ^a^	390.78 ± 10.53 ^a^	3.09 × 10^−5^
TWP	35.43 ± 1.31 ^a^	35.83 ± 1.31 ^a^	35.56 ± 1.96 ^a^	35.11 ± 1.23 ^a^	4.43 × 10^−6^
WW	155.97 ± 22.91 ^a^	163.31 ± 25.20 ^a^	168.15 ± 28.16 ^a^	146.32 ± 6.41 ^a^	2.23 × 10^−4^
WWP	13.14 ± 0.91 ^a^	13.42 ± 0.93 ^a^	13.20 ± 0.34 ^a^	13.15 ± 0.67 ^a^	0.0021

^1^ Traits associated with BLOCK3. ^2^ Displaying mean ± SD of traits by diplotype of BLOCK3. ^3^ Showing the *p* value of association between traits and BLOCK3. *p* values less than 0.001 are shown in scientific notation. ^a^ Values within a row with different superscripts differ significantly at *p* < 0.05.

## Data Availability

The raw data supporting the conclusions of this article will be made available by the authors upon request.

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
