# Peer review of "VNN1 Gene Expression and Polymorphisms Associated with Chicken Carcass Traits"

_animals, 2024, doi:10.3390/ani14131888_

Round 1
Reviewer 1 Report
Comments and Suggestions for Authors
This study provides a clear and concise overview of the possible role of VNN1 in carcass traits. Therefore, this study is interesting. However, addressing the weaknesses and clarifying the missing details would strengthen the manuscript and ensure its clarity and completeness.
Comments
1- in the abstract: Mention whether the study has considered the possible limitations of using a single chicken breed (Mahuang) for the generalizability of the results.
2- Lines 31-32: The text mentions the "yellow-feather broiler industry" but doesn't explain its significance or why it is specifically mentioned.
3- In the introduction: the specific carcass traits being analyzed are not explicitly stated.
4- The text indicates that further studies are needed to confirm the relationship between VNN1 expression with carcass traits.
5- Line 70: Age of chicken at slaughter?.
6- Line 70: The text mentions "stepped cages" but doesn't explain their specific design or purpose.
7- some grammatical errors for example:
- In line 66 (e.g. "All the chickens were kept in the brooder house").
-In line 73 "eviscerated with giblet (EWG)" to "eviscerated weight with giblets (EWG)".
8- Line 85: volume of blood collected for each bird?.
9- Line 90: size of the amplified VNN1 fragments ?.
10- In the section (2.4. RNA Extraction, Complementary DNA (cDNA) Synthesis, and Quantitative Real-Time 94 PCR (qRT-PCR)):
- number of replicates used for qRT-PCR analysis for each tissue and gene ?.
- Amount of RNA was used in the qRT-PCR for each sample ?.
- How was the random selection of 47 Mahuang chickens for liver tissue collection done?
- Did you perform any quality control (QUALITY AND QUNNTITY) on the RNA samples before cDNA synthesis?
11- In the section (2.5.):
- SNP selection or filtering (How?).
- many SNPs were ultimately analyzed (How?).
- correction for multiple testing in the association analysis. why?.
- How was the quality of the sequencing data assessed?
- Did you perform a validation of the identified associations between SNPs and carcass traits?
12- Line 135: The sentence "AFW and AFWP were significantly negative correlated to carcass traits expect for AFW and AFWP themself" is need corect.
13 - In section (3.1.): statistical tests (e.g. t-tests, ANOVAs) (add !).
14 - In section (3.1.): sex differences when analyzing carcass traits. why?.
15- Lines 159-162: the sentence "Although there may be several regulatory interventions between expression levels and phenotype, the correlation coefficients (r2) of AFW and AFWP phenotype with VNN1 expression levels were 0.2337 and 0.1961, respectively (Figure 2F and G)" is not clear.
16 - Line 160: the significance level (p-value) for the correlations was determined. How?.
17 - Line 174: It is unclear how the SNPs were identified and genotyped.
18- No validation of the newly identified SNPs is mentioned in the text.
19- In section (3.4.): "ANOVA was performed to estimate differences in AFW between genotypes at rs736763713. Chickens with the AA genotype had significantly higher AFW compared to those with GA or GG genotypes (p < 0.05, File S1)." (Rewrite)
20- Center text in all tables.
21- Line 246: Consider changing the sentence "In China, the traditional breeding focus on live poultry sales is no longer meets the evolving market demands for centralized slaughtering and chilled listing." to something like: "The traditional focus on live poultry sales in China is no longer aligned with the evolving market demand for centralized slaughtering and chilled chicken products." Please rephrase.
22- In the discussion: It should be clarified whether this study took into account the potential limitations of using a single chicken breed (Mahuang) to publish the results.
23- In the Discussion: are there plans for further studies to confirm the functional role of VNN1 in regulating abdominal fat deposition in chickens?
24- In Conclusions: The authors should slightly revise the conclusion section to be more clear.
25- References NO. 2, 3, 11, 14, 15, 17, 18, 19, 22, 23, 24, 25, 26: Abbreviation of the name of the journal.
Author Response
June 2024
Dear Reviewer,
Thank you for reviewing our manuscript (animals-3049924). We appreciate all the efforts from you to improve the manuscript. We have revised the manuscript and addressed the raised issues by you accordingly. We hope you can reconsider our revised manuscript following the comments. All corrections are highlighted in yellow color in the revised manuscript.
Comments 1: in the abstract: Mention whether the study has considered the possible limitations of using a single chicken breed (Mahuang) for the generalizability of the results.
Response 1: We appreciate the reviewer for this comment. Our primary focus was on the Mahuang chicken, which is a dominant breed in the yellow-feathered broiler market in China. We have revised the abstract to highlight our focused on Mahuang chickens. We have added the limitations of the research results and the potential for wider application in the discussion section. Revised line 291-299.
Comments 2: Lines 31-32: The text mentions the "yellow-feather broiler industry" but doesn't explain its significance or why it is specifically mentioned.
Response 2: We have added more context to explain the significance of the yellow-feather broiler industry in China. In 2023, yellow-feathered broilers accounted for 34.06% of the total number of broilers slaughtered and 25.21% of the total meat yield by broilers. We have added this information to highlight the importance of this industry and the need for improved production performance.
Comments 3: In the introduction: the specific carcass traits being analyzed are not explicitly stated.
Response 3: Thank you for comments. The carcass traits being analyzed have been stated in the section of introduction. Revised line37-41
Comments 4: The text indicates that further studies are needed to confirm the relationship between VNN1 expression with carcass traits.
Response 4: Thank you for the comment. Our literature survey revealed that VNN1 is able to regulate hepatic metabolism, which led us to speculate that hepatic expression of VNN1 might be associated with carcass traits. To confirm this relationship, we designed experiments specifically to test our hypothesis.
Comments 5: Line 70: Age of chicken at slaughter?
Response 5: Thank you for your comment. We have clarified that all Mahuang chickens were kept in stepped cages until 90 days of age and were slaughtered at 90 days of age.
Comments 6: Line 70: The text mentions "stepped cages" but doesn't explain their specific design or purpose.

Response 6: Thank you for comments. The stepped cage is a kind of cage for raising chicken. Step cages usually have three tiers, each staggered from the other, with a conveyor belt for transporting chicken manure laid at the bottom of the cage (as shown in the figure below). We have added references in text.
Comments 7: some grammatical errors for example:
- In line 66 (e.g. "All the chickens were kept in the brooder house").
Response 7: We were really sorry for our careless mistakes. We have corrected this to "All the chickens were kept in a brooder house."
-In line 73 "eviscerated with giblet (EWG)" to "eviscerated weight with giblets (EWG)".
Response 7: We were really sorry for our careless mistakes. "eviscerated with giblet (EWG)" has been corrected to "eviscerated weight with giblets (EWG)".
Comments 8: Line 85: volume of blood collected for each bird?.
Response 8: Thank you for comments. 2 ml blood was collected from each chicken.
Comments 9: Line 90: size of the amplified VNN1 fragments ?.
Response 9: Thank you for comments. We have added the size information in Table 1 in the text.
Comments 10: In the section (2.4. RNA Extraction, Complementary DNA (cDNA) Synthesis, and Quantitative Real-Time 94 PCR (qRT-PCR)):
- number of replicates used for qRT-PCR analysis for each tissue and gene ?.
Response 10: Thank you for comments. We done 2 technology replicates for each sample and gene.
- Amount of RNA was used in the qRT-PCR for each sample?.
Response 10: Thank you for comments. We used about 800 ng RNA for cDNA synthesis and 4.2 μl from 200 μl cDNA for one qRT-PCR reaction, therefore about 17 ng RNA was used for each sample per reaction.
- How was the random selection of 47 Mahuang chickens for liver tissue collection done?
Response 10: During slaughtering, many skilled workers split the chickens and we have difficulty in collecting the livers from all the chickens, so we collected one after another and would have missed most of samples, eventually collecting 47 livers. The samples obtained were random throughout the process.
- Did you perform any quality control (QUALITY AND QUNNTITY) on the RNA samples before cDNA synthesis?
Response 10: Thank you for comments. We performed quality control by using NanodropOne, and all RNA samples were over 1000 ng/μl and free of organic contamination.
Comments 11: In the section (2.5.):
- SNP selection or filtering (How?).
Response 11: Thank you for comments. Just as description in "Statistics and Sequence Analysis", we diagnosed the peak of every base to genotype possible SNP site (as shown in the figure below). Considering that we used Sanger sequencing to manually genotype avoiding the problem of a large number of missing at SNP loci, and that, in addition, our samples were from breeding populations for which it was acceptable not to conform to Hardy Weinberg equilibrium, we ended up using all the screened SNPs for association analyses.

- many SNPs were ultimately analyzed (How?).
Response 11: Thank you for comments. In this article, we focused on SNPs that may affect the expression of VNN1 and therefore screened upstream and downstream of the VNN1 gene body, resulting in 39 SNPs, which we associated with the carcass traits and haplotype analyzed the SNPs to determine if they were linked as a way of finding SNP markers associated with the carcass traits.
- correction for multiple testing in the association analysis. why?.
Response 11: Thank you for comments. We discarded correction considering 2 main factors: 1. We only tested 39 SNPs; 2. These SNPs were within one gene. We set a significance threshold of 0.05, meaning that the number of false positives among the 39 SNPs would be fewer than 2. It is noteworthy that these SNPs are located upstream and downstream of a single gene, indicating the presence of linkage disequilibrium. This linkage effect can effectively reduce the risk of Type I errors. If we were to apply a correction and set the significance threshold to 0.05/39, there would be a high probability of committing Type II errors. Therefore, we did not apply any correction.
- How was the quality of the sequencing data assessed?
Response 11: Thank you for comments. We First, we filtered out reads with a total sequencing read length of shorter than 700 bp. we then opened the ".ab1" file for each read to ensure that the chromatogram was low background, with peak heights of 500 or more and peak masses of 50 or more for most bases. Finally, we trimmed the first 30 bp and last 30 bp of the read.
- Did you perform a validation of the identified associations between SNPs and carcass traits?
Response 11: Thank you for comments. We performed an association analysis using an Affymetrix 600 K genotyping array previously created by our lab (Xu et al. 2016). We filtered out the SNPs within VNN1, resulting in the identification of 9 SNPs. Following the association analysis (as described in lines 137-143), we found that 4 SNPs were significantly associated with DW, EW, EWG, BMW, TW, and AFW. Specifically, SNP rs314708583 was found to be associated with DW, EW, EWG, BMW, and AFW (Figure 4).
|
600K |
Chr: bp |
SNP name |
Conseq.Type |
DW |
EW |
EWG |
BMW |
TW |
AFW |
|
AX-76508333 |
56890257 |
rs16283703 |
upstream |
0.3447 |
0.3151 |
0.3594 |
0.1426 |
0.5357 |
0.2214 |
|
AX-80736650 |
56894155 |
rs313420265 |
intron |
0.3219 |
0.2376 |
0.2497 |
0.4008 |
0.66 |
0.1261 |
|
AX-76508343 |
56895704 |
rs316013393 |
missense |
0.03246 |
0.01952 |
0.03057 |
0.01733 |
0.1498 |
0.01328 |
|
AX-76508344 |
56895855 |
rs313557781 |
synonymous |
0.04172 |
0.0251 |
0.0358 |
0.03441 |
0.0807 |
0.004523 |
|
AX-80924283 |
56897369 |
rs316931663 |
intron |
0.9446 |
0.6964 |
0.7484 |
0.2102 |
0.8982 |
0.3224 |
|
AX-76508350 |
56898680 |
rs315384943 |
3 prime UTR |
0.8095 |
0.796 |
0.8089 |
0.4508 |
0.761 |
0.9422 |
|
AX-76508366 |
56904356 |
rs312890334 |
intron |
0.9087 |
0.708 |
0.6761 |
0.8028 |
0.3642 |
0.3375 |
|
AX-76508369 |
56906198 |
- |
intron |
0.004033 |
0.003394 |
0.00537 |
0.002416 |
0.004296 |
0.002853 |
|
AX-76508372 |
56907337 |
rs314708583 |
3 prime UTR |
0.03919 |
0.04263 |
0.04175 |
0.04521 |
0.02592 |
0.06741 |
Comments 12: Line 135: The sentence "AFW and AFWP were significantly negative correlated to carcass traits expect for AFW and AFWP themself" is need correct.
Response 12: Thank you for comments. We have corrected it.
Comments 13: - In section (3.1.): statistical tests (e.g. t-tests, ANOVAs) (add !).
Response 12: Thank you for comments. We added t-tests for sex differences in carcass traits in 3.1.
Comments 14: In section (3.1.): sex differences when analyzing carcass traits. why?.
Response 14: Thank you for comments. The carcass traits of male and female obeyed different normal distributions, so we performed separate correlation analyses, and these results are displayed in a figure (Figure 1), where blue represents the results for males, red for females, and grey for the whole population.
Comments 15: Lines 159-162: the sentence "Although there may be several regulatory interventions between expression levels and phenotype, the correlation coefficients (r2) of AFW and AFWP phenotype with VNN1 expression levels were 0.2337 and 0.1961, respectively (Figure 2F and G)" is not clear.
Response 15: Thank you for comments. We have corrected it.
Comments 16: Line 160: the significance level (p-value) for the correlations was determined. How?.
Response 16: Thank you for comments. As we described in "2.6. Statistics and Sequence Analysis", we fitted a linear fit to the VNN1 expression and carcass traits, and in doing so we obtained a goodness-of-fit p-value by Graphpad prism 9.5.
Comments 17: Line 174: It is unclear how the SNPs were identified and genotyped.
Response 17: Thank you for comments. We have added the method for SNP identifying and genotyping in "2.5. Sequencing and Genotyping". We searched through the peaks at each base to find the location of the SNP and genotyped based on the colour of the peaks.

Comments 18: No validation of the newly identified SNPs is mentioned in the text.
Response 18: Thank you for comments. We used Sanger sequencing to discover SNPs, Sanger sequencing is the "gold standard" for detecting SNPs and the SNP detection rate for Sanger sequencing is close to 100%, so Sanger sequencing ensures that the SNPs we obtain are accurate. "Chan EY. Next-Generation Sequencing Methods: Impact of Sequencing Accuracy on SNP Discovery. In: Komar AA, ed. Single Nucleotide Polymorphisms: Methods and Protocols. Humana Press; 2009:95-111. doi:10.1007/978-1-60327-411-1_5"

Comments 19: In section (3.4.): "ANOVA was performed to estimate differences in AFW between genotypes at rs736763713. Chickens with the AA genotype had significantly higher AFW compared to those with GA or GG genotypes (p < 0.05, File S1)." (Rewrite)
Response 19: Thank you for comments. We have rewritten it. Line215-217
Comments 20: Center text in all tables.
Response 20: Thank you for comments. We have centered all the text in tables.
Comments 21: Line 246: Consider changing the sentence "In China, the traditional breeding focus on live poultry sales is no longer meets the evolving market demands for centralized slaughtering and chilled listing." to something like: "The traditional focus on live poultry sales in China is no longer aligned with the evolving market demand for centralized slaughtering and chilled chicken products." Please rephrase.
Response 21: Thank you for comments. We have rewritten it.
Comments 22: In the discussion: It should be clarified whether this study took into account the potential limitations of using a single chicken breed (Mahuang) to publish the results.
Response 21: We have to admit that a single breed (Mahuang) is indeed the limitation of our application of this study to a wide range of breeds. In this study, we took more attention to Mahuang chicken, as this breed is the dominant hot seller in the yellow-feathered broiler market in China. We have added the limitations of the research results and the potential for wider application in the discussion section.
Comments 23: In the Discussion: are there plans for further studies to confirm the functional role of VNN1 in regulating abdominal fat deposition in chickens?
Response 21: Thank you for comments. We have done some experiments to investigate the functional role of VNN1 in regulation abdominal fat deposition. In liver, we found that VNN1 responses to diet change to regulate steroid biosynthesis (peer review). In abdominal fat, VNN1 expression is upregulated during fasting, which then promote lipolysis and inhibit preadipocyte proliferation and differentiation (manuscript writing). In addition, hepatic VNN1 is a major source of serum Vanin-1, and we are exploring the role of Vanin-1 as a signal for metabolic regulation of muscle and fat (experimenting).
Comments 24: In Conclusions: The authors should slightly revise the conclusion section to be more clear.
Response 24: Thank you for comments. We have rewritten it. Line302-304
Comments 25: References NO. 2, 3, 11, 14, 15, 17, 18, 19, 22, 23, 24, 25, 26: Abbreviation of the name of the journal.
Response 25: Thank you for comments. We have updated right abbreviation of journal names.

Reviewer 2 Report
Comments and Suggestions for Authors
The article has significant practical value.
The title fully responds to the content of the article
Abstract summarize the article and points out clues
line 32 please support with statistical data; more broad presentation of the industy is needed - it will underline the value of the research
line 51 - in the opinion of reviewer comparison of humans and broilers is not a good idea
the Introduction needs re-writting, current version is quite plain and erratic
lines 64-70 please provide more data about the chickens, details of feeding, initial body weight, health check, etc
line 64 the gruop was having similar genetic line? or divers?
2.3 fresh blood was tested? no storage?
line 96 not clear number of samples
line 102 "different tissues" - please be more specific; pooled samples?
line 130 sex is not a variable value?
Tables clear, all supporting the article
discussion must be re-written, very brief, lack of broad facing research results with actual literature
conclusions too-far reaching, not possible to evaluate current breding proggrams on the base of the article
list of references limited
Comments on the Quality of English Languageno serious language faults, smoth reading possible
Author Response
June 2024
Dear Reviewer,
We feel great thanks for your professional review work on our manuscript (animals-3049924). As you are concerned, there are several problems that need to be addressed. According to your nice suggestions, we have made extensive corrections to our previous draft, the detailed corrections are listed below. All corrections are also highlighted in yellow color in the revised manuscript.
Comments 1: The article has significant practical value.
Response 1: Thank you for recognizing the practical value of our manuscript.
Comments 2: The title fully responds to the content of the article
Response 2: Thank you for your positive feedback on the title.
Comments 3: Abstract summarize the article and points out clues
Response 3: Thank you for your positive feedback on the abstract.
Comments 4: line 32 please support with statistical data; more broad presentation of the industry is needed - it will underline the value of the research
Response 4: We have added statistical data about the yellow-feather chicken industry to provide a broader context and underline the value of the research.
Comments 5: line 51 - in the opinion of reviewer comparison of humans and broilers is not a good idea
Response 5: We have removed the comparison between humans and broilers as per your suggestion.
Comments 6: the Introduction needs re-writing, current version is quite plain and erratic
Response 6: We have re-written the introduction to make it more comprehensive and coherent. Revised line 31-66
Comments 7: lines 64-70 please provide more data about the chickens, details of feeding, initial body weight, health check, etc.
Response 7: We understand the importance of this information. We have added these details in lines 72-82
Comments 8: line 64 the group was having similar genetic line? or divers?
Response 8: The 432 Mahuang chickens were from the same breeding line.
Comments 9: 2.3 fresh blood was tested? no storage?
Response 9: Yes, we collected fresh blood during slaughter using EDTA tubes and then stored them at -20°C.
Comments 10: line 96 not clear number of samples
Response 10: We apologize for the oversight. A total of 47 chicken liver samples were used for RNA extraction.
Comments 11: line 102 "different tissues" - please be more specific; pooled samples?
Response 11: We have removed the results about VNN1 expression in different tissues, as these have been previously published in "MiR-122 targets the vanin 1 gene to regulate its expression in chickens." The section has been corrected accordingly.
Comments 12: line 130 sex is not a variable value?
Response 12: We agree with this suggestion and have now included separate counts for carcass traits of females and males.
Comments 13: Tables clear, all supporting the article
Response 13: Thank you for your affirmation.
Comments 14: discussion must be re-written, very brief, lack of broad facing research results with actual literature
Response 14: We have re-written the discussion to include a broader range of research results and references to actual literature. Revised line265-304
Comments 15: conclusions too-far reaching, not possible to evaluate current breeding programs on the base of the article
Response 15: We have revised the conclusions to make them more grounded and aligned with the data presented in the article.
Comments 16: list of references limited
Response 16: We have expanded the list of references to include more research results and literature that support our study.

Round 2
Reviewer 1 Report
Comments and Suggestions for Authors
Apparently the authors have responded to all comments and inquiries. So in my opinion it is acceptable in animals.
Reviewer 2 Report
Comments and Suggestions for Authors
Previous suggestions of the reviewer were taken into consideration by the Author.
tha manuscript is improved